# Low-temperature grapho-epitaxial La-substituted BiFeO$_3$ on metallic perovskite

Sajid Husain [1,8] ✉, Isaac Harris [2,8], Guanhui Gao[3], Xinyan Li [3], Peter Meisenheimer [4], Chuqiao Shi[3], Pravin Kavle [1,4], Chi Hun Choi [3], Tae Yeon Kim[4], Deokyoung Kang [4], Piush Behera[1,4], Didier Perrodin[1], Hua Guo[3], James M. Tour [3,5], Yimo Han [3], Lane W. Martin [1,4,7], Zhi Yao [6] ✉ & Ramamoorthy Ramesh [1,2,4,7] ✉

Bismuth ferrite has garnered considerable attention as a promising candidate for magnetoelectric spin-orbit coupled logic-in-memory. As model systems, epitaxial BiFeO$_3$ thin films have typically been deposited at relatively high temperatures (650–800 °C), higher than allowed for direct integration with silicon-CMOS platforms. Here, we circumvent this problem by growing lanthanum-substituted BiFeO$_3$ at 450 °C (which is reasonably compatible with silicon-CMOS integration) on epitaxial BaPb$_{0.75}$Bi$_{0.25}$O$_3$ electrodes. Notwithstanding the large lattice mismatch between the La-BiFeO$_3$, BaPb$_{0.75}$Bi$_{0.25}$O$_3$, and SrTiO$_3$ (001) substrates, all the layers in the heterostructures are well ordered with a [001] texture. Polarization mapping using atomic resolution STEM imaging and vector mapping established the short-range polarization ordering in the low temperature grown La-BiFeO$_3$. Current-voltage, pulsed-switching, fatigue, and retention measurements follow the characteristic behavior of high-temperature grown La-BiFeO$_3$, where SrRuO$_3$ typically serves as the metallic electrode. These results provide a possible route for realizing epitaxial multiferroics on complex-oxide buffer layers at low temperatures and opens the door for potential silicon-CMOS integration.

Multiferroic materials possess multiple coexisting order parameters[1], allowing for the effective control of (anti)ferromagnetic order with an electric field or magnetic field control of electrical polarization[2–5]. These properties make them desirable for applications in magnetic field sensing and novel magnetic memory designs for ultrafast and efficient nonvolatile, in-memory computing. One such design, the so-called magnetoelectric spin–orbit logic device, makes use of multiferroics along with a spin–orbit-coupled material for efficient reading and writing of a nanomagnetic bit[6]. The perovskite bismuth ferrite

(BiFeO$_3$, BFO)−with a robust spontaneous polarization (~1 C/m$^2$ aligned along the pseudocubic <111> direction) and canted antiferromagnetic ($L$) order, is one of the most exciting multiferroics due to its high ferroelectric Curie (1143 K) and Néel (643 K) temperatures[7]. Furthermore, the electrically insulating nature of BFO offers a promising path for magnonic devices, where the transfer of spin via magnons plays a key role in energy-efficient and fast information processing[8]. To tune the multiferroic properties of thin-film BFO such as coercive field, magnetic and polar order,

[1]Materials Sciences Division, Lawrence Berkeley National Laboratory, Berkeley, CA 94720, USA. [2]Department of Physics, University of California, Berkeley, CA 94720, USA. [3]Department of Materials Science and NanoEngineering, Rice University, Houston, TX 77005, USA. [4]Department of Materials Science and Engineering, University of California, Berkeley, CA 94720, USA. [5]Department of Chemistry, Rice University, Houston, TX 77005, USA. [6]Applied Mathematics and Computational Research Division, Lawrence Berkeley National Laboratory, Berkeley, CA 94720, USA. [7]Present address: Department of Physics and Astronomy, Rice University, Houston, TX 77005, USA. [8]These authors contributed equally: Sajid Husain, Isaac Harris. ✉e-mail: shusain@lbl.gov; jackie_zhiyao@lbl.gov; rr73@rice.edu

rare-earth-cation doping (e.g., with lanthanum (La) as in $Bi_{1-x}La_xFeO_3$, BLFO) is typically used[9].

Along with cation doping, the choice of substrate and bottom electrode also has a significant impact on the multiferroic properties of BFO thin films. Epitaxial BFO and BLFO thin films are typically grown on metallic perovskites (e.g., $SrRuO_3$, SRO) at high temperatures (650–800 °C), and on lattice-matched substrates (e.g., $DyScO_3$, $TbScO_3$ and $SrTiO_3$, STO)[1–3,5,8,10–13]. For device integration, however, it is essential to select a substrate and growth temperature that is compatible with silicon-CMOS processing, while maintaining the desired properties of the B(L)FO films and the bottom electrode.[14] It is known that STO can be deposited epitaxially on silicon substrates[15], however, the B(L)FO growth temperatures of 600–800 °C are incompatible with traditional CMOS processing, which typically requires fabrication temperatures of no more than 450 °C[14,16] to avoid damage to underlying CMOS components. The lattice mismatch with CMOS materials leads to challenges in fabrication processes, which require efforts to process the epitaxial BFO growth on very different lattice structures. This may involve the use of buffer layers (primarily discussed in this communication), specialized fabrication techniques (such as pulsed-laser deposition), and innovations in material engineering (e.g., elemental substitutions). Several attempts have been made to grow BFO at low temperatures, but the resulting films have ended up being polycrystalline and exhibit diminished ferroic properties[17–19].

Here, we demonstrate highly [001] textured/epitaxial growth of $Bi_{0.85}La_{0.15}FeO_3$ (BLFO) thin film on the metallic perovskite $BaPb_{0.75}Bi_{0.25}O_3$ (BPBO) at a substrate heater temperature as low as 450 °C on STO (001) substrates. This approach addresses major concerns of compatibility with CMOS processing and shows promise for practical applications. From a technological perspective, we believe the total concentration of lead should be well below what is the allowable limit; the mass of Pb on 6” CMOS wafer (500 μm) is calculated to be 0.02 ppm, which is under the permissible limit (<0.1% or 1000 ppm)[20] (Frequently asked questions about Toshiba semiconductor company's RoHS-compatible(*) and 'Lead(Pb)-Free'(**) semiconductors and support for customers converting to RoHS-compatible manufacturing.). The electrode system, BPBO, is known for its historical significance as one of the first high transition temperature oxide superconductors prior to the discovery of the cuprates and exhibits a superconducting transition temperature of ~11 K (ref. 21) at a composition $x = 0.25$. Its normal, metallic state at room temperature is characterized by a resistivity of ~ 2 mΩ•cm (ref. 22). Recent work on BPBO has shown that in addition to superconductivity, BPBO has a strong spin-Hall effect[22], making it attractive for potential use in spintronic devices. This work, which addresses the low-temperature synthesis requirements for CMOS compatibility and combines the promising transport physics of the BPBO electrode with the robust ferroelectricity in BLFO and opens the door to testing and implementation of multiferroics integrated into CMOS devices. Specifically, this work investigates and compares all-epitaxial heterostructures of BPBO/BLFO/BPBO which were grown using pulsed-laser deposition (PLD) at 450 °C, to standard PLD deposited SRO/BLFO/SRO grown at high temperatures (700 °C) and SRO/BLFO/SRO heterostructures grown at low temperature (450 °C). (Supplementary Fig. 1 for detailed temperature/pressure-dependent samples.) Particularly, the role of the bottom electrode as a template that structurally and chemically seeds to perovskite BFO (or BLFO) layer is noteworthy. X-ray diffraction (XRD), reciprocal space mapping (RSM), and high-resolution cross-sectional transmission electron microscopy (TEM) are used to characterize epitaxial nature and the ferroelectric polarization vector mapping. Piezoresponse force microscopy (PFM) and capacitor-based polarization electric field hysteresis loops (and related switching studies) were used to characterize the ferroelectric properties of the films at room temperature.

## Results

The PLD growth of BPBO and BLFO at 450 °C has been optimized ("Methods"), resulting in symmetric, highly textured/epitaxial trilayer. The crystalline quality of the BPBO/BLFO/BPBO trilayers is compared to standard high-temperature-grown SRO/BLFO/SRO as well as SRO/BLFO/SRO heterostructures grown at 450 °C using XRD and RSM studies ("Methods"). While BPBO and BLFO both share a perovskite lattice structure, there is a ~9% lattice mismatch between BPBO and BLFO and, as such, one would not expect any templating of the [001] orientation of the BLFO layer on such a surface. Surprisingly, it is found that all layers grow "cube-on-cube" (Fig. 1a) with the [001] of all lattices oriented normal to the plane of the substrate and single phase even when grown at 450 °C. This is enabled by the low-temperature epitaxial growth of the BPBO electrode. On the other hand, the SRO quality is severely affected at 450 °C, which leads to poor BLFO growth. This is demonstrated by RSM studies; the 450 °C BPBO peak is sharp and shows thickness oscillations indicating high-quality growth, whereas the 450 °C SRO peak is diffuse indicating poor coherence, and the resulting BLFO peak on 450 °C SRO is much more diffuse than the BLFO peak on 450 °C BPBO. We have attempted BLFO deposition (on BPBO) at higher temperatures but noticed that the heterostructure starts to become unstable at 550 °C (Supplementary Fig. 1a). Thus, the BLFO epitaxy is set by the BPBO layer which is grown epitaxially at 450 °C, therefore the lowest temperature that we have been able to use for BLFO growth is 450 °C. From the 2θ-ω diffraction results (Fig. 1b), a single phase corresponding to the out-of-plane lattice constant ($c$) for BLFO and BPBO are calculated to be 3.96 Å and 4.29 Å, respectively. The similar values in and out-of-plane lattice constant of BPBO (Table 1) proves the BPBO grown fully relaxed on STO substrate consistent with the domain-matching epitaxy reported by others[23–26]. The lattice constant of BLFO grown at 450 °C is slightly smaller than the bulk value of BFO[27] due to the lanthanum substitution but larger than that of the strained BLFO ($a = 3.91$ Å, $c = 4.00$ Å) grown at 700 °C on SRO electrodes. This suggests that the BPBO and BLFO are fully relaxed on the STO substrates, a point further evidenced by the differences in parallel wavevectors in RSM studies between the substrate, BLFO, and BPBO peaks. The lattice parameters of low-temperature-grown BPBO and BLFO are close to their bulk values[28], which suggests relaxed, highly oriented growth. In contrast, the high-temperature-grown SRO/BLFO/SRO heterostructures (Fig. 1d) are fully strained to the substrate. On the other hand, the low-temperature-grown SRO/BLFO/SRO heterostructures (Fig. 1e) also shows relaxed BLFO and SRO layers and it appears that the BLFO has deteriorated. The differences in strain states results in differences in the lattice parameters of the materials, and the in-plane $a \left( = \frac{\lambda\sqrt{(h^2+k^2)}}{2Q_x} \right)$ and out-of-plane $c \left( = \frac{\lambda l}{2Q_z} \right)$ lattice parameters (calculated using RSM) can be compared (Table 1). Here $h$, $k$, and $l$ are the indices and $\lambda$, $Q_x$, and $Q_z$ correspond to the wavelength of the X-ray and the parallel and perpendicular wavevectors, respectively.

From symmetric and asymmetric X-ray scans, we conclude that the BLFO grows with a "cube-on-cube" orientation despite the large mismatch of ~9% with the BPBO underlayer. Here, we briefly examine the potential model involved in relaxed epitaxial heterostructure growth in situations where the lattice mismatch is significantly large. Lattice-matched epitaxy refers to the condition where the film maintains a fixed orientation relative to the substrate or underlayer, defining the epitaxial relationship. It is established that a film-substrate lattice difference of greater than 7% (up to 22% or in some cases over 30%[29]) leads to film growth via grapho-epitaxy[30] or also known as domain-matching epitaxy[26], where the matching is achieved in multiples of lattice translations in the two layers. Thin-film epitaxial growth is typically constrained by three key factors: (1) the lattice mismatch between the substrate and film (referred to as lattice misfit), (2) variations in the thermal expansion coefficients of the different materials

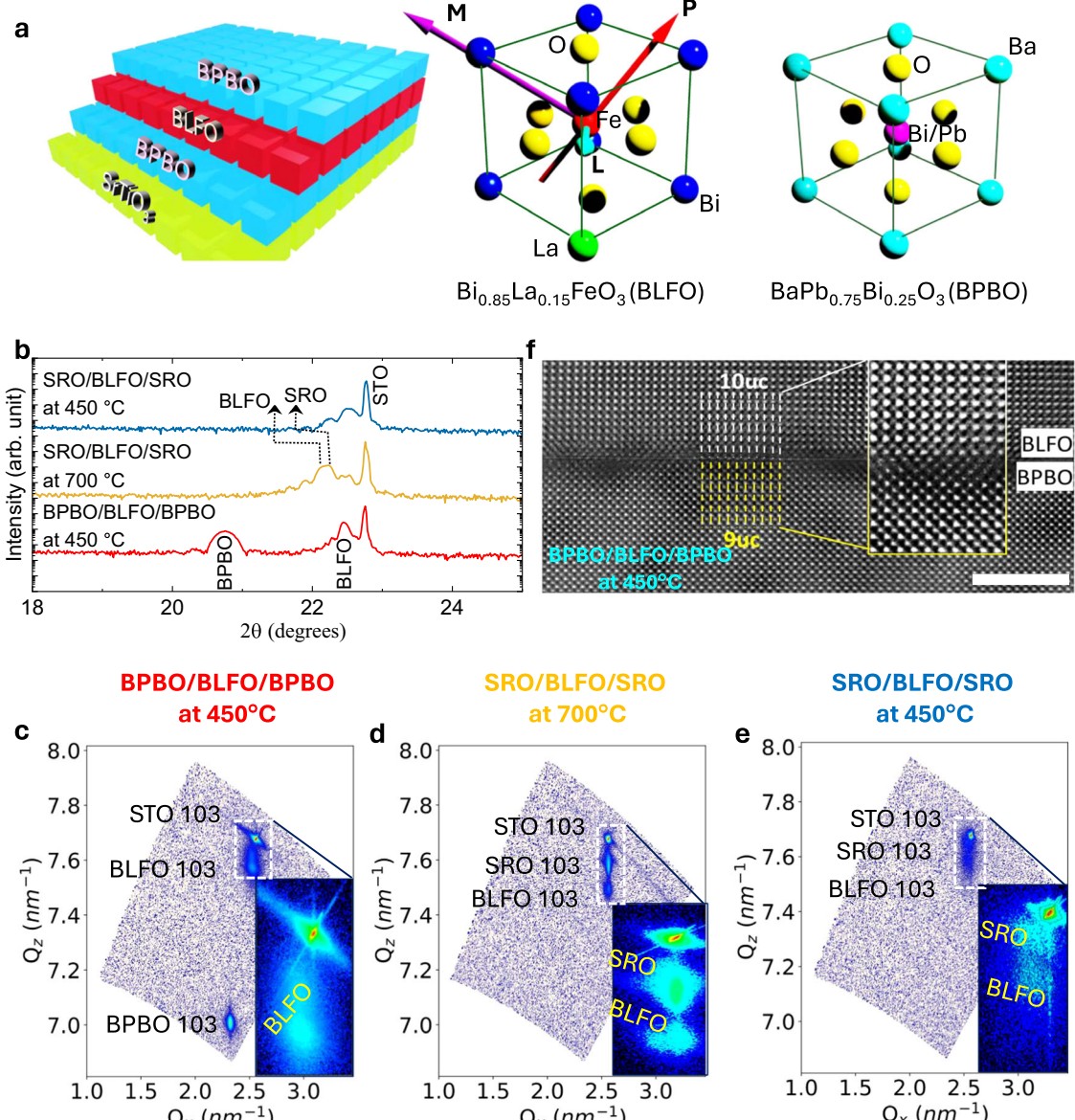

**Fig. 1 | Structural characteristics. a** Schematic depicting the cube-on-cube growth of BPBO/BLFO/BPBO on STO substrate. Unit cells of BLFO (arrows represent the different order parameters such as polarization 'P', magnetic 'M', and anti-ferromagnetic 'L' vector) and $BaPb_{0.75}Bi_{0.25}O_3$ (BPBO). **b** 2θ-ω X-ray diffraction patterns of BPBO and SRO based heterostructures prepared in different temperatures. Reciprocal space mapping of samples **c** BPBO/BLFO/BPBO grown at 450 °C, **d** SRO/BLFO/SRO grown at 700 °C, and **e** SRO/BLFO/SRO grown at 450 °C. RSM mapping corresponding to (103) asymmetric plane of STO. The corresponding BLFO and BPBO 103-diffraction peaks appear at different $Q_x$ and $Q_z$ vectors depending on the in-plane and out-of-plane lattice parameters. Insets are zoomed cross-section of BLFO and SRO for temperature-dependent heterostructures. Atomic imaging: **f** Cross-sectional HAADF-STEM image of BLFO/BPBO deposited at 450 °C. The magnified HAADF-STEM image, and (inset) reveals 10uc/9uc domain epitaxy between BLFO and BPBO. The scale bar is 10 Å.

**Table 1 | In-plane (*a*) and out-of-plane (*c*) lattice parameters of BLFO, SRO and BPBO evaluated using RSM data**

| Stack ↓ / Parameters → | BLFO | | SRO | | BPBO | |
|---|---|---|---|---|---|---|
| | *a* (Å) | *c* (Å) | *a* (Å) | *c* (Å) | *a* (Å) | *c* (Å) |
| BPBO/BLFO/BPBO at 450 °C | 3.96 | 3.96 | – | – | 4.31 | 4.29 |
| SRO/BLFO/SRO at 700 °C | 3.91 | 4.00 | 3.91 | 3.96 | – | – |
| SRO/BLFO/SRO at 450 °C | 3.95 | 3.95 | 3.93 | 3.92 | – | – |

(i.e., thermal expansion mismatch), and (3) the microstructural strains due to the defects or substitution dopants.

The thermal expansion coefficients of BFO ($10–14 × 10^{-6}$/K)[31], $BaPbO_3$ ($10–15 × 10^{-6}$/K)[32] and STO substrate ($9 × 10^{-6}$/K) (Crystec-GmbH) are similar; therefore, it is not expected that thermal expansion mismatch contributes significantly to the strain. Thus, in the domain-matching epitaxy paradigm (for a system of lattice misfit ~9%), 10-unit cells of BLFO and 9-unit cells of BPBO are expected to be accommodated with ~1% of mismatch. High-angle annular dark-field scanning transmission electron microscopy (HAADF-STEM) images (Fig. 1f) confirm the epitaxial growth of BLFO on BPBO as described above based on the domain-matching epitaxy. Based on this analogy, BLFO is energetically relaxed on top of the BPBO and expected to have

minimal strain at the interface as observed in RSM studies. The question arises as to why then the BLFO does not exhibit the same growth behavior on SRO as it does on BPBO (as seen at low temperature, Fig. 1e) since the thermal expansion coefficient of SRO ($11 \times 10^{-6}$/K)[33] is of the same order of magnitude to BFO and the lattice mismatch is relatively small. Note that from the data (Fig. 1d, e), the SRO grows epitaxially at 700 °C, but its quality is significantly compromised when deposited at 450 °C, indicating the loss of crystalline ordering during low-temperature growth. In contrast, BPBO exhibits single crystalline epitaxial growth at 450 °C, indicating that the crucial factor is the electrode layer, which needs to be epitaxial and highly crystalline at the desired temperature to facilitate the epitaxial growth of BLFO. It may be noted that the Pb−O terminated surface provides low surface energy ($\sim 1$ eV/nm²)[34] for adatoms and potentially helps in the nucleation process to grow epitaxial BLFO at sufficiently low temperature in comparison to the high surface energy of Sr−O surface ($\sim 6$ eV/nm²)[35]. The BLFO growth on STO at 450 °C temperature (Supplementary Fig. 2) shows the polycrystalline nature of BLFO. This suggests the importance of epitaxial BPBO having low surface energy elements such as Pb−O for progressive epitaxial thin-film growth. Formation of dislocations at the interface, however, cannot be avoided because of strain relaxation, which may affect the electronic (and dielectric/ferroelectric) properties of the materials. The STEM images of the controlled samples deposited at 700 °C and 450 °C on SRO electrode are shown in Supplementary Fig. 3. The atomic-scale imaging is consistent with the RSM analysis where the SRO/BLFO/SRO heterostructure deposited at 700 °C is well-ordered while deteriorated at 450 °C.

To understand the effect of growth temperature on the polar state of BLFO, we have performed a detailed analysis of HAADF-STEM image vector maps and then calculated the atomic displacement from them. Although such STEM images only provide a measure of the structural distortion and the broken inversion symmetry in the material, one can infer the degree of polar order from such data (common perovskite ferroelectrics such as BaTiO₃, PbTiO₃, and BFO all exhibit a direct relationship between the spontaneous dipole moment and the structure distortion)[36–38]. The Fourier-filtered HAADF-STEM images were analyzed (Methods) using CalAtom Software to extract the atomic position of La/Bi and Fe ions by multiple-ellipse fitting[39]. The Fe displacement vector in each unit cell was calculated by confirming the center of mass of its four closest La/Bi neighbors. Quantitatively, the polarization displacement vector analysis is shown in Fig. 2. High/low-temperature deposited BLFO atomic image and P-vector map (Fig. 2a–d) reveals the polarization distribution among the 1000's of unit cells. The displacement vector distribution profiles at three different places shown in Fig. 2e indicate a non-uniform polar displacement in different regions. What is perhaps most important is that although the material is still polar, the degree of long-range order within the BLFO layer is corrupted (Supplementary Figs. 4–8). In contrast, the high-temperature BLFO shows uniform polarization displacement distribution (Fig. 2a, c and Supplementary Figs. 4–8) consistent with our previous report[9]. It is noteworthy that the issue of long-range order of the polar state in such robust ferroelectrics has received very little attention, in contrast to the extremely well-studied relaxor ferroelectrics[38,40,41]. For instance, what is the extent of long-range coherency of the order parameter that is required for the material to exhibit the full value of the spontaneous polarization? We elaborate more on this after the discussion of the ferroelectric measurements.

Invigorated with the understanding of the structure of the BLFO grown at low temperature, we now turn to ferroelectric measurements. In order to probe how the degree of long-range structural order impacts the macroscopic polarization/dielectric/piezoelectric responses, we first demonstrate the piezoelectric and ferroelectric behavior in the BLFO films using piezoresponse force microscopy (PFM). To do this, films were deposited without the top BPBO electrode, and the PFM tip is used as the "top electrode" for application of localized electric field. To simultaneously demonstrate the piezoelectric and ferroelectric nature, an out-of-plane "box-in-a-box" pattern was written with the PFM tip by applying −5 V to a 7-μm square,

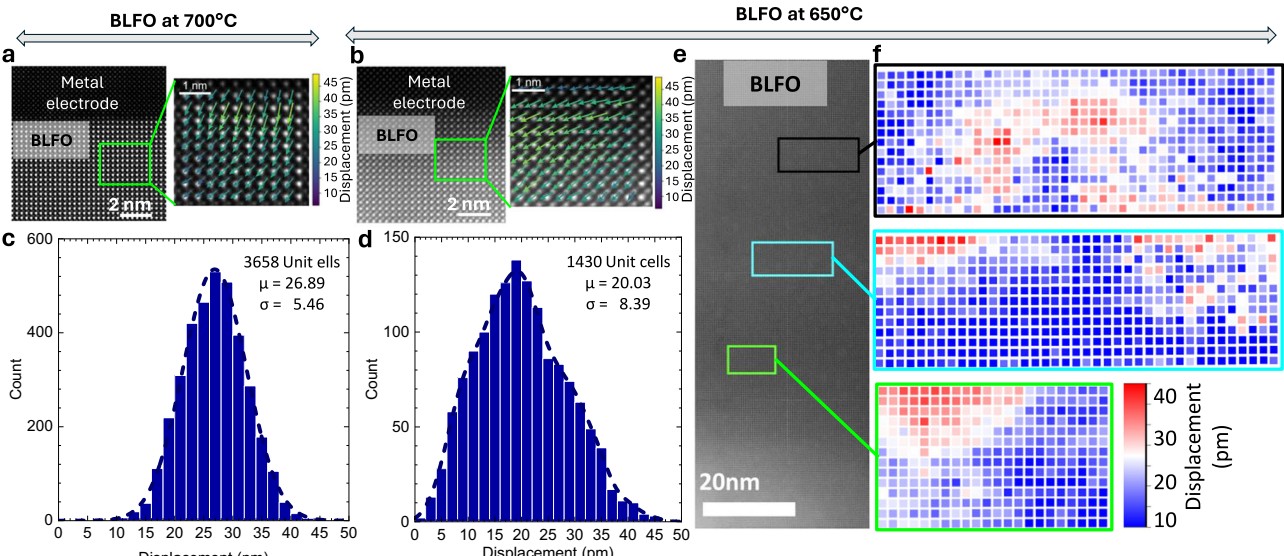

**Fig. 2 | Atomic imaging and polarization mapping.** Polarization displacement vector mapping on BLFO atomic images collected on the BLFO thin films cross-section deposited at **a** 700 °C and **b** 450 °C. Color arrows indicate the magnitude of polarization mapping vis-à-vis atomic displacement. Polarization displacement has been calculated using the Fourier-filtered HAADF-STEM images and the histograms includes the data corresponding to 3658- and 1430-unit cells of BLFO, respectively, for the sample deposited at 700 °C (**c**) and 450 °C (**d**). μ is the center of the histogram, or the mean value of atomic displacement and σ is the standard deviation. The dotted line is the smoothening curve. High-temperature BLFO follows the Gaussian distribution, whereas the low-temperature sample shows non-uniform distribution of polarization displacement vector. **e** HAADF-STEM image of BLFO grown on BPBO at 450 °C. **f** Represent the corresponding polarization displacement distribution maps across the area of the image at different places. Zoomed TEM images and corresponding polarization displacement distribution plots are represented in Supplementary Fig 8.

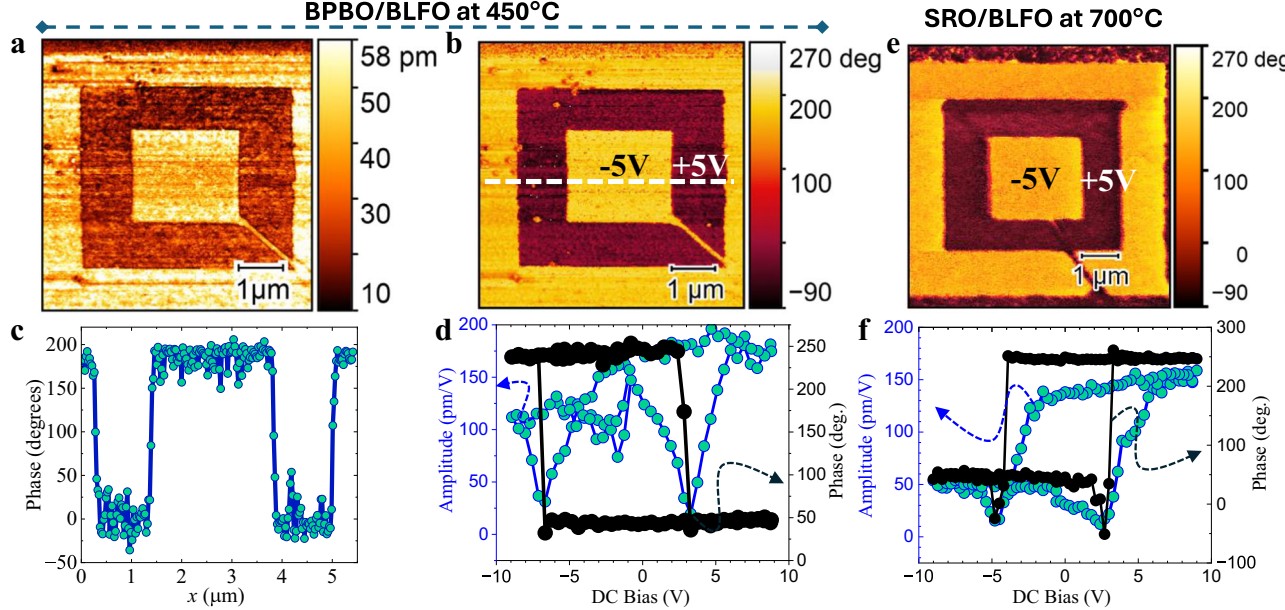

**Fig. 3 | Piezoelectric response. a** Piezoelectric amplitude and **b** piezoresponse force microscopy image of BPBO/BLFO. Box-in-box mapping is done with ±5 V electric field. **c** Line scan profile of the BLFO phase of (**b**). **d** Point-contact amplitude and piezoelectric hysteresis. **e, f** PFM and piezoelectric hysteresis data for SRO/BLFO deposited at 700 °C. Both samples loops were recorded at a frequency 0.83 Hz and 1 V AC drive amplitude with ±9 V DC scan voltage.

followed by +5 V to a 5-µm square inside the first, and then −5 V to a 2.5 µm square inside the second, with all squares being concentric. The resulting domain structures are then imaged through the amplitude and phase of the cantilever response in Fig. 3a, b, respectively. The light (dark) regions correspond to the net polarization vector pointing out of the plane of the film (into the plane of the film), and the sharp contrast in the images as well as the clear 180° phase switch in the line profile (Fig. 3c) indicate consistent switching between two stable out-of-plane directions. Furthermore, the retention of the written "box-in-a-box" pattern was probed and found to be stable even after 24 h (Supplementary Fig. 9), and we anticipate such performance to remain stable for even longer durations. PFM images recorded using different applied voltages are also provided (Supplementary Fig. 10) and reveal the critical local-switching voltage. Finally, by leaving the tip in a fixed spot on the sample and measuring the amplitude and phase upon application of a ± 9 V triangular wave (frequency 0.83 Hz), a characteristic square hysteresis loop in the phase and butterfly loops in the amplitude, corresponding to changes in the polarization and piezoelectric amplitude, respectively, are observed (Fig. 3d); demonstrating polarization reversal driven by the applied voltage. Similarly, PFM image and the hysteresis were recorded on SRO/BLFO sample deposited at 700 °C ("Methods"). The coercive field and piezo amplitude are found to be relatively larger in low-temperature BPBO/BLFO sample. The large piezo amplitude in low-temperature BPBO/BLFO is indicative of the enhanced dielectric behavior (discussed later). The local-ferroelectric loops are shifted horizontally due to the different conductivities of the platinum PFM probe tip and bottom (BPBO/SRO) electrode. No PFM contrast is observed in the 450 °C SRO/BLFO sample (Supplementary Fig. 11), suggesting minimal piezoelectricity in this sample, which suggests that the BLFO in this case is not ferroelectric (or weakly so).

Having confirmed the ferroelectric behavior of BLFO by PFM, we proceeded to quantitatively investigate the ferroelectric properties in macroscopic capacitors. Polarization electric field hysteresis loops (Fig. 4a) demonstrate robust ferroelectricity in the 450 °C BPBO/BLFO/BPBO (heterostructure I; Fig. 4a) and the 700 °C SRO/LBFO/SRO (heterostructure II; Fig. 4a) heterostructures, but not in the 450 °C SRO/BLFO/SRO (heterostructure III; Fig. 4a) heterostructures. Notably,

there are two important differences between heterostructures I and II. First, the absolute magnitude of the switched polarization in heterostructure I is ~50% that of heterostructure II. Microscopically, this can be traced back to the lower degree of long-range order in the films grown at 450 °C. The second key difference is that the coercive voltage for heterostructure I is ~2× that of heterostructure II. On the surface, this appears counter-intuitive, given that the spontaneous polarization is smaller by ~50%. Thus, the lower degree of long-range order in heterostructure I not only reduces the spontaneous polarization but makes switching of the polar state a lot more energetically costly.

Our first hypothesis for this difference was that the BLFO deposited at 450 °C is likely to have a lower oxygen content and a larger degree of cationic disorder. To examine the role of oxygen, we post-annealed heterostructure I in oxygen at atmospheric pressure. The polarization hysteresis of BPBO/BLFO/BPBO heterostructures deposited at 450 °C and ex situ annealed (for comparison) at different temperatures (Supplementary Fig. 12) indicates robust ferroelectricity in BLFO; however, the remnant polarization is lower and the coercive field is higher than the high-temperature SRO based heterostructures. As discussed by HAAD-STEM polarization mapping, the short-range order of polarization vector is responsible for the reduction the remnant polarization and enhances the coercive field[37,38]. Moreover, the sheet resistance (1.2 kΩ/sq.) of BPBO is larger than that of SRO (0.58 kΩ/sq.), which possibly contribute to an additional source of voltage drop across the capacitor that enhances the coercive fields[42]. This further opens the possibility of tuning the BPBO conductivity through different doping and future aspects of the low-temperature BFO. We also expect the band inversion because of different work functions of (n-type) SRO and (p-type) BPBO (Supplementary Note 2; Hall effect measurements to identify the carrier type in SRO thin film and energy band schematics of BPBO and SRO based heterostructures) where the charge transport across the capacitor is governed by electrons and holes in SRO and BPBO, respectively; however, this should not affect the intrinsic coercivity. The characteristic dielectric permittivity-electric field hysteresis (Fig. 4b, measured at 50 kHz ac oscillations and amplitudes of 100 mV), confirm that the 700 °C SRO/BLFO/SRO and the 450 °C BPBO/BLFO/BPBO heterostructures are sufficiently insulating (capacitive) to give rise to

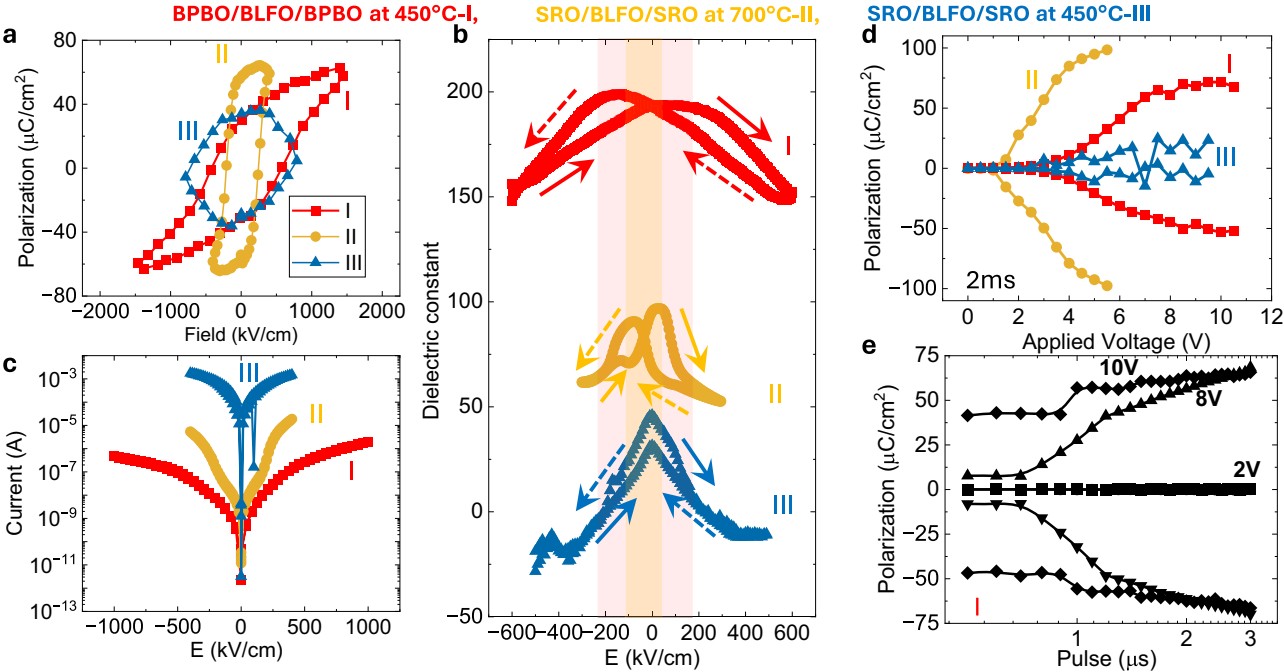

**Fig. 4 | Ferroelectric response. a** Polarization as function of electric field for the samples BPBO/BLFO/BPBO at 450 °C→I, SRO/BLFO/SRO at 700 °C→II, and SRO/BLFO/SRO at 450 °C→III. **b** dielectric constant, and **c** Leakage current as a function of electric field across the capacitors in BPBO/BLFO/BPBO and SRO/BLFO/SRO heterostructures. **d** Pulsed switched polarization as a function of varying voltage at a fixed 2 ms pulse width. **e** Voltage dependence of switched polarization by varying the pulsed widths in sample-I.

dielectric responses, whereas the 450 °C SRO/BLFO/SRO heterostructures appear to be highly leaky and likely non-ferroelectric. As discussed above, the piezoelectric hysteresis (Fig. 3d, f) shows a relatively larger piezo amplitude suggests the higher value of dielectric constant in low-temperature BLFO. The dielectric constant is also a function of oxygen defect concentration, domain wall boundaries or electrically inhomogeneous microstructure. The low-temperature BLFO is expected to have some of these effects dominate, which potentially could contribute to a larger dielectric constant. This is further supported by the high leakage currents observed (Fig. 4c) for the 450 °C SRO/BLFO/SRO heterostructures compared to the others, as well as the low resistance (~35 kΩ) of the capacitors probed. The broader peaks observed in the XRD and RSM of the 450 °C SRO/BLFO/SRO sample suggest a lower degree of ordering and the presence of a higher density of defects/dislocations (Supplementary Fig. 2). This may explain the higher leakage currents observed (Fig. 4b) compared to the 700 °C SRO/BLFO/SRO heterostructures, as dislocations/defects/vacancies can serve as channels for leakage currents in ferroelectrics[43]. This is expected due to the deteriorated SRO growth (which serves as a template for BLFO deposition) at low temperatures as also previously reported by prior works[44–46].

Further, we performed the PUND (positive up negative down) switching measurements at constant pulse-width varying voltage (Fig. 4d) and constant–voltage varying pulse width (Fig. 4e). It can be seen again that there is switching in both heterostructures I and II, but not in the low-temperature SRO based BLFO (heterostructure III). The polarization switching voltages are higher in the 450 °C BPBO/BLFO/BPBO (I) heterostructure than the 700 °C SRO/BLFO/SRO (II). Lower value of polarization in 450 °C BLFO is consistent with the analysis of HAADF-STEM polarization mapping. As explained above, the electrical properties of the electrode may play an important role in deciding the voltage distribution and screening effect across the electrode interfaces[42,47]. In our case, the BPBO has low carrier concentration as well as higher resistivity that potentially serve as an additional voltage drop and impact the switching voltage, however, these parameters

(such as electrode conductivity and carrier mobility) can be further improved by tuning the Bi/Pb substitution[48] in BPBO lattice though the epitaxial quality should not compromise at low temperature.

Finally, we carried out fatigue and retention measurements on these BPBO/BLFO/BPBO and SRO/BLFO/SRO heterostructures grown at 450 °C and 700 °C temperatures, respectively. For comparison, we also performed ex situ post-annealing on the BPBO/BLFO/BPBO heterostructures at 450 °C (same as the deposition temperature) and 500 °C for 2 h. The flat lines in the retention measurements (Fig. 5a) show that the heterostructures retain their polarization state as a function of time ($10^6$ s -10 days). Bipolar fatigue studies reveal a minimal decrease in the polarization after $10^7$–$10^8$ cycles (Fig. 5b) of the 450 °C BPBO/BLFO/BPBO heterostructures. This problem is alleviated by ex situ annealing the 450 °C BPBO/BLFO/BPBO heterostructures in oxygen at 450 °C and 500 °C for 2 h (Fig. 5b). Furthermore, these results reveal a measurably higher polarization value for the annealed samples, indicating that oxygen vacancies contribute not only to the initial value of the polarization but also to the fatigue behavior. This finding is consistent with the growth optimization, where changes in oxygen pressure during deposition and in the annealing/cool-down steps have substantial impact on the crystallinity and ferroelectric properties.

In summary, we demonstrate a method for growing BLFO at low temperatures on metallic BPBO electrodes. Despite a 9% lattice mismatch, the oriented growth of the ferroelectric phase is achieved at temperatures as low as 450 °C, due to the high-quality low-temperature epitaxial BPBO. Piezoelectric force microscopy and measurements on macroscopic capacitors demonstrate ferroelectric properties comparable to high-temperature BLFO grown on standard SRO electrodes. We note that existing work on single-crystal BLFO and BFO is done at deposition temperatures that are significantly higher and thus less compatible with CMOS device integration[14], and therefore this work presents a new avenue for integration of the multiferroic in CMOS devices. We also show that the low-temperature BLFO has lost long-range polarization ordering which results in the lower remnant

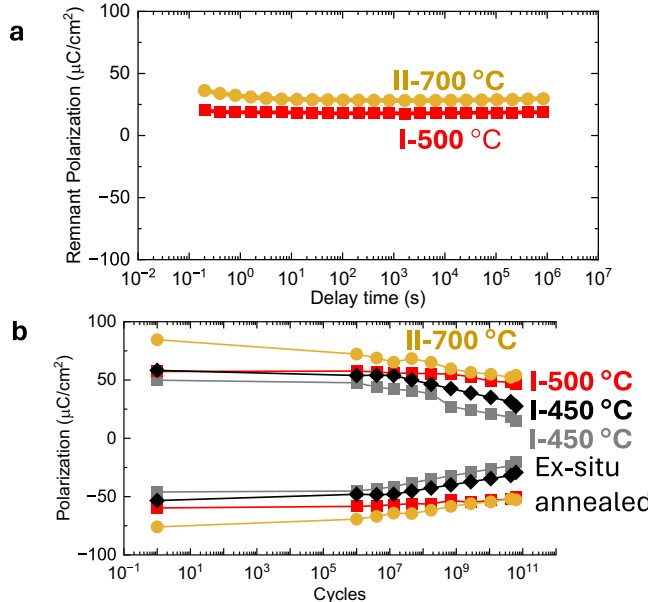

**Fig. 5 | Retention and fatigue. a** Retention and **b** fatigue tests in BPBO/BLFO/BPBO and SRO/BLFO/SRO heterostructures I after different temperature treatments. High-temperature BLFO heterostructure II deposited at 700 °C is also shown for comparison.

polarization. This may stem from the relaxed nature of BLFO on BPBO template. Due to the conducting nature of BPBO and sharp interfaces of all epitaxial heterostructure, the electrical properties are consistent even full BLFO stack is deposited at a record low temperature. Our study provides a new pathway to control the temperature of multiferroic thin-film growth as well as the polarization ordering, which help in further consideration of the BFO for a range of new applications.

## Methods

### Pulsed-laser deposition of thin-film heterostructures
All thin films are prepared by PLD using a 248 nm KrF excimer laser (COMPex-Pro, Coherent) in an on-axis geometry with a target-substrate distance of 5 cm. We clean (100) STO substrates (Krystec) by sonication in Acetone and Isopropyl Alcohol for 5 min each and fix our substrates to a heater using thermally conductive silver paint. Our BPBO target is prepared by mixing powders of BaO, $Pb_3O_4$, and $Bi_2O_3$ to achieve 10% excess of Pb and Bi in the stoichiometry for $x = 0.25$, pressing the mixture at 20 kPSI, and sintering at 850 °C for 10 h. Our $BaPb_{1-x}Bi_xO_3$ (BPBO) target is prepared by mixing and grinding powders of BaO, $Pb_3O_4$, and $Bi_2O_3$ in an agate mortar (4 N purity) to achieve 10% excess of Pb and Bi in the stoichiometry for $x = 0.25$. The mixture is pressed for the first time at 10 kPSI for 15 min in a 3/8" steel die to form a pellet and then uniformly pressed at 30kPSI for 30 min in an AIP CP360 isostatic press. The pellet is finally sintered and at 750 °C for 12 h followed by a 48 h slow cooling to prevent any cracking. The obtained target has a slightly lower diameter than the initial pellet due to its densification. For low-temperature BLFO depositions, we use a stoichiometric $Bi_{0.85}La_{0.15}FeO_3$ target (Praxair "Linde"). For BPBO deposition, we achieve a base pressure of $5 \times 10^{-6}$ Torr, raise the substrate temperature to 450 °C, introduce a dynamic $O_2$ environment of 100 mTorr, and pulse the laser on the target with 5 Hz repetition rate at a fluence of 0.13 J/cm². This first BPBO layer is then annealed in situ at 435 °C for 1 h under a static $O_2$ environment of 400 torr. The temperature is then reset to 450 °C and the BLFO layer is grown with a laser fluence of 1.5 J/cm² in a dynamic oxygen environment of 50 mTorr with 1 Hz repetition rate. Finally, the top BPBO electrode is grown in the same manner as the bottom electrode, and the heterostructure is cooled to room temperature at 30 °C/min in a static $O_2$ environment of

500 Torr. For high- and low-temperature SRO depositions, we use a dynamic $O_2$ environment of 100 mTorr, pulse rate of 5 Hz, fluence of 0.8 J/cm², stoichiometric target (Praxair "Linde"), and temperature of 700 °C and 450 °C, respectively. For the high-temperature BLFO deposition, we use 110 mTorr dynamic $O_2$ environment, 10 Hz pulse rate, 1.8 J/cm² fluence, and a target with 18% excess Bi (Praxair 'Linde'). Some samples were grown without the top electrode for PFM. Controlled BLFO samples were also deposited directly on STO substrate. For carrier concentration measurements of SRO, controlled sample of 75-nm-thick SRO film was deposited on STO(001) substrate at 700 °C, 200 mTorr, 15 Hz, and 1.8 J/m².

### Crystal structure determination
The crystal lattice parameters of all layers are measured by XRD and RSM (asymmetric 2D scan) using a high-resolution X-ray diffractometer (PANalytical, X'Pert MRD) with a Cu $K_\alpha$ ($\lambda = 1.5401$ Å) source and a PIXcel3D-Medipix3 detector. For RSM, the diffraction along the STO (103) plane is used, which allows to gain the information along the [103] direction of BPBO and BFO. Film thicknesses and deposition rates are extracted from the spacing of the thickness fringes in the XRD as well as from oscillations in X-ray reflectometry.

### Cross-section sample preparation and high-angle annular dark-field scanning transmission electron microscopy (HAADF-STEM).
The cross-section samples were prepared using a Helios660 scanning electron microscope/focused ion beam (SEM/FIB) with a gallium (Ga) ion beam source. After sample preparation, the cross-section samples were analyzed using a FEI Titan Themis G3 scanning/transmission electron microscope (S/TEM) equipped with double correctors and monochromator. The microscope was operated under high-angle annular dark-field scanning transmission electron microscopy (HAADF-STEM) mode, utilizing 300 kV accelerating voltage. In HAADF-STEM mode, a 25 mrad convergence angle was employed for both imaging and energy-dispersive X-ray (EDX) spectrum mapping. For imaging purposes, a spot size of eight was used. For energy-dispersive X-ray (EDX) mapping, a spot size of six was used.

### Quantification and mapping of atomic displacement and lattice parameters
Fourier-filtered HAADF-STEM images were analyzed using CalAtom software to extract the atomic position of Bi/La and Fe ions by multiple-ellipse fitting[39]. The Fe displacement vector in each unit cell was calculated by confirming the center of mass of its four closest Bi/La neighbors. The displacement vector **D** of the Fe column is represented as follows:

$$\boldsymbol{D} = \boldsymbol{r_{Fe}} - \frac{\boldsymbol{r_1} + \boldsymbol{r_2} + \boldsymbol{r_3} + \boldsymbol{r_4}}{4},$$

where $\boldsymbol{r_{Fe}}$ is the position vector of the Fe column. $\boldsymbol{r_1}, \boldsymbol{r_2}, \boldsymbol{r_3}, \boldsymbol{r_4}$ are the position vectors of the four closest Bi/La neighbors in each unit cell. The color of the displacement vectors was represented by the vector magnitude. The displacement histograms were calculated by the magnitude of the displacement vector in each unit cell. As for BLFO deposited at 700 °C, the histogram was collected within Supplementary Fig. 7a; the histogram of BLFO at 450 °C was collected within three regions of Supplementary Fig. 8b.

The lattice parameters were also calculated unit cell-by-unit cell as follows:

$$\Delta_{x,i} = x_i - x_{i-1}$$

$$\Delta_{y,i} = y_i - y_{i-1}$$

where $\Delta_{x,i}$ and $\Delta_{y,i}$ are the lattice parameters of the $i$th unit cell. $x_i$ and $y_i$ are the coordinates of the $i$th Bi/La column. The visualization of the two-dimensional atomic displacement and lattice parameter mapping was carried out using Python.

## Ferroelectric domain imaging

Piezoelectric force microscopy (PFM) imaging is done using the MFP-3D, Asylum Research AFM. All measurements use a silicon cantilever coated with Pt (Kurt J. Lesker) as the conducting top electrode for local application of an electric field.

For the correctness of PFM phase and amplitude hysteresis, the data was recorded at different places to rule out non-uniformity in the sample. We also observed that a brand-new PFM tip can only be used for two scans for the best comparison of the piezoelectric amplitude. We have taken care to try to eliminate all the possible measurement artifacts to help enable a quantitative comparison between two samples.

## Macroscopic capacitor fabrication

The ferroelectric characterization of the heterostructures is done with 40-nm-thick BPBO (or 50-nm-thick SRO) films serving as the top and bottom electrodes, surrounding 100 nm BLFO. The measurements are performed on circular capacitors with a diameter of 25 μm. To fabricate devices, we first deposit Pt(5 nm) on the trilayer heterostructures to protect the top BPBO layer using a room-temperature DC magnetron sputtering system. We then spin coat with photoresist and do optical lithography using a mask to make a circular pattern on the heterostructures. Using an ion mill (Intlvac), we etch away the top electrode and stop upon etching of the BLFO film using a secondary ion mass spectrometer (SIMS, Hidden Analytical). Finally, we rinse off the photoresist hard mask with sonication in Acetone and Isopropyl alcohol.

We measured the resistivity of BPBO and found it to be comparable to the reported bulk value. The sheet resistance of our BPBO electrodes is ~1.2 kΩ/sq., and the vertical resistance of the whole capacitor stack (top to bottom electrode for a 25 μm diameter capacitor) is ~40 MΩ.

## Ferroelectric hysteresis and dielectric characterization

Polarization as a function of the electric field across macroscopic capacitors is measured using a Precision Fast Hysteresis (100 V ferroelectric tester from Radiant Technologies, Inc.) at room temperature. Dielectric measurements are performed using an E4990A Impedance Analyzer (Keysight Technologies) at room temperature.

## Reporting summary

Further information on research design is available in the Nature Portfolio Reporting Summary linked to this article.

# Data availability

The data that support the findings of this study are available from the corresponding author upon reasonable request.

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

## Acknowledgements

S.H. would like to thank Abel Fernandez for valuable discussions. This work was primarily supported by the U.S. Department of Energy, Office of Science, Office of Basic Energy Sciences, the Microelectronics Co-Design Research Program, under contract no. DE-AC02-05-CH11231 (Codesign of Ultra-Low-Voltage Beyond CMOS Microelectronics) for the development of novel materials for low-power microelectronics. P.K. acknowledges support from the Intel Corporation as part of the COFEEE program. T.K. acknowledges support from the Army Research Office under grant W911NF-21-1-0126. L.W.M. and R.R. also acknowledge partial support from the Army/ARL as part of the Collaborative for Hierarchical Agile and Responsive Materials (CHARM) under cooperative agreement W911NF-19-2-0119. X.L., C.H.C., and Y.H. are supported by the Welch Foundation (C-2065-20210327). Y.H. acknowledges the support from NSF (CMMI-2239545). We acknowledge the Electron Microscopy Center, Rice.

## Author contributions

R.R. and S.H. conceived the idea. S.H. and I.H. performed thin-film growth, device fabrication, and measurements. G.G., X.L., C.S., C.H.C., and H.G. performed the cross-sectional measurements and polarization mapping under the supervision of J.M.T. and Y.H. P.K. helped in reciprocal space mapping. P.M. P.B. and P.K. helped with the ferroelectric measurements. D.P. prepared the $BaPbBiO_3$ target. T.Y.K. and D.K. helped in controlled sample preparation and measurements. S.H. and I.H. wrote the manuscript with the help of R.R., Z.Y., and L.W.M. R.R. and Z.Y. supervised the work. All authors have participated in the discussion and helped in the analysis.

## Competing interests

The authors declare no competing interests.
