## [Peer Review File · Nature Communications]

Low-temperature grapho-epitaxial La-substituted BiFeO₃ on a metallic perovskiteReviewers' Comments:

Reviewer #1:

Remarks to the Author:

Review on: Low-temperature grapho-epitaxial BiFeO₃ on a bismuth- substituted metallic perovskite Husain et al report on the low-temperature epitaxial growth of lanthanum-substitutes BiFeO₃ on BPBO epitaxial layers, employing pulsed laser deposition technique. The authors demonstrated the quality of the epitaxial layers using various techniques in this manuscript. While the essence of the work is interesting, I would like to share a few reservations about publishing the work in Nature Communications in the present form. Please see the following comments:

1. It's not clear from the manuscript what is the role of Lanthanum in epitaxial growth of BFO on SRO or BLPO. The title of the manuscript is grapho-epitaxial BiFeO₃, while it is Bi_{0.85}La_{0.15}FeO₃ to be precise. Also, the presence of Pb in BLPO is of some concern, while the entire technology is going to be Lead-free.
2. While emphasizing the CMOS-compatibility of process temperature, it would be important to know at what is the minimum temperature required for the epitaxial growth. Would BLFO grow epitaxially on BPBO at lower temperatures?
3. The X-ray diffraction peaks in Figure 1 (b) show the BLFO (written 'LBFO' in the figure) peaks in 700oC-deposited SRO/BLFO/SRO stack at around 22o, while the BLFO peaks in 450o samples are around 22.5o. Is this due to the strain in the lattice? It is mentioned on page 7, line 161 that the SRO/BLFO/SRO is well ordered at 700oC and is deteriorated at 450oC. XRD of BPBO/BLFO/BPBO looks similar to SRO/BLFO/SRO at 450oC. (For a proper comparison between heterolayers, a comparison between BPBO/BLFO/BPBO deposited at 450oC and 700oC will be necessary).
4. Figure 2 c & d are intriguing. BLFO deposited at 700oC shows a cleaner Gaussian distribution, while BLFO deposited at 450oC shows a non-uniform distribution of the polarization displacement vector, meaning that the high-temperature BLFO is better in its uniformity (which is evident from comparing Fig. 1 c & d).
5. The nonuniformity in the polarization vector of 450oC-deposited BLPO is reflected in Figure 4 (a), where the polarization measurements are shown (that has been stated on lines 287-288). While the reason for lower polarization in low-temperature BLPO is the short-range polarization, assigning the reason for this to the carrier densities in BPBO and BLFO does not make sense. The authors explicitly mention that the polarization of BPBO/BLFO/BPBO is a lot more energetically costly (lines 237-240).
6. Figure 4 b and c: what is the film thickness of BLFO in these three heterostructures? While the polarization vectors in 450oC BLFO are less oriented, what would be the reason for the higher dielectric constant of this compared to BLFO deposited at 700oC?
7. Lines 331-334: two statements are there on the preparation of BLPO targets.

Reviewer #2:

Remarks to the Author:

The authors provide a possible route for realing epitaxial multiferroics on complex-oxide buffer layer at low temperatures. The paper is well organized with sufficient references. It will be useful for researchers in the field of multiferroics thin film grown and the design of ferroelectric integrated devices. Therefore, I would like to recommend its publication in Nature Communications after the authors address the following comments.

1. Why the light regions in Figure 3 correspond to the net polarization vector pointing out of the plane after applying positive tip voltage? Should it point into the plane of the film?

2. During the epitaxial growth process, whether the mismatch between BFPO and STO is considered.
3. I propose the authors should add more discussions about the relationship between this work and silicon-CMOS integration.

Reviewer #3:

Remarks to the Author:

The manuscript entitled "Low-temperature grapho-epitaxial BiFeO₃ on a bismuth-substituted metallic perovskite" by Husain et al. describes a method for growing La-doped BiFeO₃ (BLFO) at low temperature of 450 °C on BaPb_{0.75}Bi_{0.25}O₃ (BPBO) metallic electrodes. The authors performed a detailed microstructural and electronic characterizations of the BLFO films on BPBO in comparison with those films on SRO electrodes deposited on various temperatures. This study may provide a possible way to grow multiferroic epitaxial films at low temperatures for integrating with conventional silicon-CMOS. The results presented in this study are interesting, but some issues should be addressed clearly.

(1) The authors attributed the oriented growth of the ferroelectric phase at low temperature to high-quality epitaxial BPBO electrode which may provide smooth surface for epitaxial growth. However, it seems difficult to grow ferroelectric BLFO epitaxial films at temperature as low as 450 °C even on atomically flat STO substrate without any buffer layers. Have authors achieved high-quality BLFO films on STO or Nb-doped STO substrate? Whether other substance like PbO with low melting point has contribution on the low temperature growth?

(2) As shown in Figure 4 and Figure 5, higher growing and ex-situ annealing temperatures can significantly improve the ferroelectricity, retention and fatigue properties of BLFO films, demonstrating structural defects or component fluctuations. These results suggest the electrical properties of the films grown at low temperatures are not good enough. So, the advantages of low temperature growth seems not outstanding.

(3) It is generally known that SRO is p-type conductor, not n-type described in Figure s7.

Response to comments from the Reviewer #1

Remarks from Reviewer: *Husain et al report on the low-temperature epitaxial growth of lanthanum-substituted BiFeO₃ on BPBO epitaxial layers, employing pulsed laser deposition technique. The authors demonstrated the quality of the epitaxial layers using various techniques in this manuscript. While the essence of the work is interesting, I would like to share a few reservations about publishing the work in Nature Communications in the present form. Please see the following comments:*

Response: We thank the reviewer for finding the work interesting and providing important suggestions/comments for improving the quality of the manuscript for publication. Here we present point-by-point responses to the comments.

Comment#1 *It's not clear from the manuscript what is the role of Lanthanum in epitaxial growth of BFO on SRO or BLFO. The title of the manuscript is grapho-epitaxial BiFeO₃, while it is Bi_{0.85}La_{0.15}FeO₃ to be precise. Also, the presence of Pb in BLFO is of some concern, while the entire technology is going to be Lead-free.*

Response: We thank the reviewer for this query. We chose the La doped BFO due to its advantage of low electric field polarization switching over BFO, in other words, it is electrically softer than BFO (mentioned in introduction). We agree with the reviewer (this was a simple oversight) and as per the suggestion we have changed **title** to “**Low-temperature grapho-epitaxial La-substituted BiFeO₃ on a metallic perovskite**”.

On adding Pb into BaBiO₃, the BaPbBiO₃ (BPBO) becomes high T_c superconductor around 10K [Ref.20]. Our ongoing project is to merge the multiferroic and superconducting interfaces to probe the low temperature electron transport. Another advantage of BPBO is exceptionally large spin-charge conversion efficiency [Ref.21], which is important to manipulate the antiferromagnetism in BLFO. From a technological perspective, we believe the total concentration of lead is well below what is the allowable limit; the mass of Pb on 6” CMOS wafer (500μm) is calculated to be 0.02ppm, which is under the permissible limit (<0.1%, <https://www.compliancegate.com/rohs-directive/> and https://www.toshiba.com/taec/environment/docs/faq_leadfree_0407.pdf).

Revision in manuscript (lines68-71) “.....with CMOS processing and shows promise for practical applications. **From a technological perspective, we believe the total concentration of lead should be well below what is the allowable limit; the mass of Pb on 6” CMOS wafer (500μm) is calculated to be 0.02ppm, which is under the permissible limit (<0.1% or 1000ppm)^{20,21}.** The electrode system, BPBO, is known for its historical

Comment#2 *While emphasizing the CMOS-compatibility of process temperature, it would be important to know at what is the minimum temperature required for the epitaxial growth. Would BLFO grow epitaxially on BPBO at lower temperatures?*

Response: The maximum allowed thermal load for CMOS integration is ~450°C which is the aim of this work (mentioned in introduction). The BLFO epitaxy is set by the BPBO layer which is grown epitaxially at 450°C, therefore the lowest temperature that we have been able to use for BLFO growth is 450°C. We have attempted higher temperatures but noticed that the heterostructure starts to become unstable at 550°C (Figure S1a). Thanks for the important query, we have extended the discussion in the revised manuscript. We note that the lowest temperature we have been able to grow BPBO is 425C. Further reduction in the BLFO growth temperature may be

possible by assisted growth processes such as plasma / ozone assisted growth to provide activated oxygen.

Revision in manuscript (lines105-112) “.....the BLFO peak on 450°C BPBO. We have attempted BLFO deposition (on BPBO) at higher temperatures but noticed that the heterostructure starts to become unstable at 550°C (Figure S1a). Thus, the BLFO epitaxy is set by the BPBO layer which is grown epitaxially at 450°C, therefore the lowest temperature that we have been able to use for BLFO growth is 450°C. From the $2\theta-\omega$ diffraction results (Figure 1b), a single phase corresponding to the out-of-plane lattice constant (c) for BLFO and BPBO are calculated to be 3.96 Å and 4.29 Å, respectively. The similar values in and out of plane lattice constant of BPBO (Table-I) proves the BPBO grown fully relaxed on STO substrate consistent with the domain matching epitaxy reported by others²⁴⁻²⁷. The lattice constant”

Comment#3 *The X-ray diffraction peaks in Figure 1 (b) show the BLFO (written ‘LBFO’ in the figure) peaks in 700oC-deposited SRO/BLFO/SRO stack at around 22°, while the BLFO peaks in 450° samples are around 22.5°. Is this due to the strain in the lattice? It is mentioned on page 7, line 161 that the SRO/BLFO/SRO is well ordered at 700°C and is deteriorated at 450°C. XRD of BPBO/BLFO/BPBO looks similar to SRO/BLFO/SRO at 450°C. (For a proper comparison between heterolayers, a comparison between BPBO/BLFO/BPBO deposited at 450°C and 700°C will be necessary).*

Response: We thank the reviewer for pointing out the mistake in naming and raising important concerns. The typo is now corrected.

Yes, the peak difference is due to the strain in the film as explained by RSM. BLFO in BPBO/BLFO/BPBO stack is relaxed (RSM Figure 1c) due to which the BLFO peak has different Q_x (in-plane lattice constant) with respect to the bottom BPBO layer and substrate STO. This gives the same lattice parameters; however, a large difference in lattice parameters is observed in SRO based 700°C sample (Table-I).

We have deposited BPBO/BLFO/BPBO heterostructures at high temperature range (Figure S1a) and it is shown that at 550 C°, BPBO does not prefer to grow epitaxial. We further extended this discussion in the revised manuscript.

Revision in manuscript (lines105-112) “.....the BLFO peak on 450°C BPBO. We have attempted BLFO deposition (on BPBO) at higher temperatures but noticed that the heterostructure starts to become unstable at 550°C (Figure S1a). Thus, the BLFO epitaxy is set by the BPBO layer which is grown epitaxially at 450°C, therefore the lowest temperature that we have been able to use for BLFO growth is 450°C. From the $2\theta-\omega$ diffraction results (Figure 1b), a single phase corresponding to the out-of-plane lattice constant (c) for BLFO and BPBO are calculated to be 3.96 Å and 4.29 Å, respectively. The similar values in and out of plane lattice constant of BPBO (Table-I) proves the BPBO grown fully relaxed on STO substrate consistent with the domain matching epitaxy reported by others²⁴⁻²⁷. The lattice constant”

Comment#4 *Figure 2 c & d are intriguing. BLFO deposited at 700°C shows a cleaner Gaussian distribution, while BLFO deposited at 450°C shows a non-uniform distribution of the polarization displacement vector, meaning that the high-temperature BLFO is better in its uniformity (which is evident from comparing Fig. 1 c & d).*

Response: We thank the reviewer for noting this. Indeed, this was precisely the reason for carrying out these studies. We agree with the reviewer that the polarization is indeed spatially

non-uniform in the low temperature BLFO as discussed in the manuscript. We note that the issue of spatial uniformity of the polar order in ferroelectrics is rarely focused on, except for the cases of relaxor ferroelectrics. In some sense, this is what motivated our detailed STEM based vector mapping studies. We will be pursuing this in even more detail in future work.

Comment#5. *The nonuniformity in the polarization vector of 450°C-deposited BLPO is reflected in Figure 4 (a), where the polarization measurements are shown (that has been stated on lines 287-288). While the reason for lower polarization in low-temperature BLPO is the short-range polarization, assigning the reason for this to the carrier densities in BPBO and BLFO does not make sense. The authors explicitly mention that the polarization of BPBO/BLFO/BPBO is a lot more energetically costly (lines 237-240).*

Response: We agree with the reviewer and the irrelevant text is now removed in the revised manuscript.

Comments#6: *Figure 4 b and c: what is the film thickness of BLFO in these three heterostructures? While the polarization vectors in 450°C BLFO are less oriented, what would be the reason for the higher dielectric constant of this compared to BLFO deposited at 700°C?*

Response: We thank the reviewer for the very important comment/suggestion.

The piezoelectric hysteresis is measured under identical conditions and the relatively larger piezo amplitude also suggests a higher value of dielectric constant in the low temperature grown BLFO.

For sanity check, the data was recorded at different places to rule out non-uniformity in the sample. We also observed that a brand-new PFM-tip can only be used for two scans for best comparison of the piezoelectric amplitude. We have taken care to try to eliminate all the possible measurement artifacts to help enable a quantitative comparison between two samples.

The dielectric constant is also a function of oxygen defect concentration, domain wall boundaries or electrically inhomogeneous microstructure. The low temperature BLFO is expected to have some of these effects dominate, which potentially could contribute to a larger dielectric constant.

We once again thank you for the very good suggestion. The revised new figure and the corresponding text are added to the main text in the revised manuscript.

Figure 3 Piezoelectric response: **a** piezoelectric amplitude and **b** piezoresponse force microscopy image of BPBO/BLFO. Box-in-box mapping is done with $\pm 5\text{V}$ electric field. **c** Line scan profile of the BLFO phase on **b**. **d** Point-contact amplitude and piezoelectric hysteresis. (**e,f**) PFM and piezoelectric hysteresis data for SRO/BLFO deposited at 700°C . **Both loops were recorded at a frequency 0.83Hz and 1V AC drive amplitude with $\pm 9\text{V}$ DC scan voltage.**

Revision in manuscript (lines 214-217) "...the applied voltage. Similarly, PFM image and the hysteresis were recorded on SRO/BLFO sample deposited at 700°C (method). The coercive field and piezo amplitude are found to be relatively larger in low temperature BPBO/BLFO sample. The large piezo amplitude in low temperature BPBO/BLFO is an indicative of the enhanced dielectric behavior (discussed later). The local-ferroelectric loops ..."

Revision in manuscript (lines 255-260) ".....highly leaky and likely non-ferroelectric. As discussed above, the piezoelectric hysteresis (Figure 3d, 3f) shows relatively larger piezo amplitude suggests the higher value of dielectric constant in low temperature BLFO. The dielectric constant is also a function of oxygen defect concentration, domain wall boundaries or electrically inhomogeneous microstructure. The low temperature BLFO is expected to have some of these effects dominate, which potentially could contribute to a larger dielectric constant. This is further supported by the high leakage currents observed (Figure...."

Comments#7: *Lines 331-334: two statements are there on the preparation of BLPO targets.*

Response: Yes, these lines belong to the preparation of BPBO target.

Response to comments from the Reviewer #2

The authors provide a possible route for realing epitaxial multiferroics on complex-oxide buffer layer at low temperatures. The paper is well organized with sufficient references. It will be useful for researchers in the field of multiferroics thin film grown and the design of ferroelectric integrated devices. Therefore, I would like to recommend its publication in Nature Communications after the authors address the following comments.

Response: We are grateful to the reviewer for recommending our work for publication. Here we present point-by-point responses to the comments.

Comment#1. *Why the light regions in Figure 3 correspond to the net polarization vector pointing out of the plane after applying positive tip voltage? Should it point into the plane of the film?*

Response: We agree with the reviewer and the mistake is now corrected in the revised manuscript.

Comment#2. *During the epitaxial growth process, whether the mismatch between BFPO and STO is considered.*

Response: The BPBO film is grown fully relaxed on STO substrate (RSM data). We thank the reviewer for important suggestions. The STO discussion is also added in the revised manuscript.

Revision in the manuscript (lines 110-112) “.....for BLFO and BPBO are calculated to be 3.96 Å and 4.29 Å, respectively. The similar values in and out of plane lattice constant of BPBO (Table-I) proves the BPBO grown fully relaxed on STO substrate consistent with the domain matching epitaxy reported by others²⁴⁻²⁷. The lattice constant”

Comment#3. I propose the authors should add more discussions about the relationship between this work and silicon-CMOS integration.

Response: We appreciate the reviewer and as per suggestions, discussion is extended in the revised manuscript.

Revision in the manuscript

Lines 58-62 “..... than 450°C^{14,16} to avoid damage to underlying CMOS components. The lattice mismatch with CMOS materials leads to challenges in fabrication processes, which require efforts to process the epitaxial BFO growth on very different lattice structures. This may involve the use of buffer layers (primarily discussed in this communication), specialized fabrication techniques (such as pulsed laser deposition), and innovations in material engineering (e.g., elemental substitutions). Several.....”

Revision in manuscript (lines 68-71) “...with CMOS processing and shows promise for practical applications. From a technological perspective, we believe the total concentration of lead should be well below what is the allowable limit; the mass of Pb on 6” CMOS wafer (500µm) is calculated to be 0.02ppm, which is under the permissible limit (<0.1% or 1000ppm)^{20,21}. The electrode system, BPBO, is known for its historical ...”

(lines 308-310-) “.....BPBO template. Due to the conducting nature of BPBO and sharp interfaces of all epitaxial heterostructure, the electrical properties are consistent even full BLFO stack is deposited at a record low temperature. Our study provides a new pathway to control the temperature of....”

Response to comments from the Reviewer #3

The manuscript entitled “Low-temperature grapho-epitaxial BiFeO₃ on a bismuth-substituted metallic perovskite” by Husain et al. describes a method for growing La-doped BiFeO₃ (BLFO) at low temperature of 450 °C on BaPb_{0.75}Bi_{0.25}O₃ (BPBO) metallic electrodes. The authors performed a detailed microstructural and electronic characterizations of the BLFO films on BPBO in comparison with those films on SRO electrodes deposited on various temperatures. This study may provide a possible way to grow multiferroic epitaxial films at low temperatures for integrating with conventional silicon-CMOS. The results presented in this study are interesting, but some issues should be addressed clearly.

Response: We are grateful to the reviewer for appreciating the work and finding the study important and interesting for integrating with conventional silicon-CMOS. Here we present point-by-point responses to the issues raised by the reviewer.

Comment#1. The authors attributed the oriented growth of the ferroelectric phase at low temperature to high-quality epitaxial BPBO electrode which may provide smooth surface for epitaxial growth. However, it seems difficult to grow ferroelectric BLFO epitaxial films at temperature as low as 450 °C even on atomically flat STO substrate without any buffer layers. Have authors achieved high-quality BLFO films on STO or Nb-doped STO substrate? Whether other substance like PbO with low melting point has contribution on the low temperature growth?

Response: We thank the reviewer for the very important question. Data from new sample of BLFO deposited directly on STO substrate at 450°C and 700°C shown below indicate that the film does not prefer to be epitaxial at low temperature on STO. A new figure (Figure S2) and corresponding text is added to the revised manuscript and supplementary.

We agree with the reviewer’s suggestion that the Pb-O (~1eV/nm²) [ref.³⁵] may help in the preferred orientation growth due to its lower surface energy than the Sr-O (~6eV/nm²) [ref.³⁶]. We agree on the point that the presence of Pb-O terminated surface provides low surface energy for adatoms and helps in the nucleation process to grow epitaxial BLFO at sufficiently low temperature. We have added this discussion in the revised manuscript.

Revision in the manuscript and supplementary

Lines 157-163 “temperature to facilitate the epitaxial growth of BLFO. It may be noted that the Pb-O terminated surface provides low surface energy (~1eV/nm²) [ref.³⁵] for adatoms and potentially helps in the nucleation process to grow epitaxial BLFO at sufficiently low temperature in comparison to the high surface energy of Sr-O surface (~6eV/nm²) [ref.³⁶]. The BLFO growth on STO at 450°C temperature (Supplementary information Figure S2) shows the polycrystalline nature of BLFO. This suggests the importance of epitaxial BPBO having low surface energy elements such as Pb-O for progressive epitaxial thin film growth. Formation of dislocations at the interface, however,”

Figure S2 | X-ray diffraction patterns of BFLO thin film deposited at 450°C and 700°C temperatures on STO (001) substrate. Epitaxial growth preferred at high temperature on STO substrate whereas the additional phase appeared at low temperature. Peak broadening and slightly redshift in the (002) peak of 450°C BLFO implies the significant lattice deformation at low temperature. * Represent the substrate peaks.

Comment#2 *As shown in Figure 4 and Figure 5, higher growing and ex-situ annealing temperatures can significantly improve the ferroelectricity, retention and fatigue properties of BLFO films, demonstrating structural defects or component fluctuations. These results suggest the electrical properties of the films grown at low temperatures are not good enough. So, the advantages of low temperature growth seems not outstanding.*

Response: We want to emphasize that proposed method is to reduce the growth temperature from 700°C to 450°C and open a pathway for further research towards possible CMOS integration.

Comment#3 *It is generally known that SRO is p-type conductor, not n-type described in Figure s7.*

Response: SRO is typically n-type in its pristine form on STO substrate Phys. Rev. Lett. 107, 127601 (2011) and Nat. Comm. 11, 184 (2020). These references are also added in the supplementary.

Reviewers' Comments:

Reviewer #1:

Remarks to the Author:

The authors have given sufficient explanations to the queries and suggestions have been considered in the revised version. I would recommend publishing this article in Nature Communications.

Reviewer #2:

Remarks to the Author:

I find the authors have addressed all the concerns raised by the reviewer, and now I recommend this paper for publication.

Reviewer #3:

Remarks to the Author:

Husain et al. have addressed some of my comments satisfactorily. Before recommending the revised manuscript for publication, the conduction type of SRO at room temperature should be made clearly otherwise it would be misleading for audiences. The authors argued SRO is n-type conductor in its pristine form on STO based on 2 references. However, there are also many literatures have reported SRO bulk exhibits positive Seebeck coefficient values in the range 160–800 K (PHYSICAL REVIEW B 91, 045106 (2015); Phys. Rev. B 73, 052412 (2006). Materials Transactions, Vol. 49, No. 3 (2008) pp. 600 ; Journal of Alloys and Compounds, 387, 56-59.) Indicating p-type conduction. The Hall effect measurements of SRO films on STO also show the slope of R_{xy} is positive when temperatures are higher than Curie temperature. (PHYSICAL REVIEW MATERIALS 4, 054414 (2020), Phys. Rev. B 103, 085121.). Because many researches demonstrate that SRO exhibits p-type conductivity, the authors should provide the Hall effect measurement of SRO film to confirm their argument.

Reviewer #1

The authors have given sufficient explanations to the queries and suggestions have been considered in the revised version. I would recommend publishing this article in Nature Communications.

Response: We thank the reviewer for recommending the manuscript for publication.

Reviewer #2

I find the authors have addressed all the concerns raised by the reviewer, and now I recommend this paper for publication.

Response: We thank the reviewer for recommending the manuscript for publication.

Reviewer #3

Husain et al. have addressed some of my comments satisfactorily. Before recommending the revised manuscript for publication, the conduction type of SRO at room temperature should be made clearly otherwise it would be misleading for audiences. The authors argued SRO is n-type conductor in its pristine form on STO based on 2 references. However, there are also many literatures have reported SRO bulk exhibits positive Seebeck coefficient values in the range 160–800 K (PHYSICAL REVIEW B 91, 045106 (2015); Phys. Rev. B 73, 052412 (2006). Materials Transactions, Vol. 49, No. 3 (2008) pp. 600 ; Journal of Alloys and Compounds, 387, 56-59.) Indicating p-type conduction. The Hall effect measurements of SRO films on STO also show the slope of R_{xy} is positive when temperatures are higher than Curie temperature. (PHYSICAL REVIEW MATERIALS 4, 054414 (2020), Phys. Rev. B 103, 085121.). Because many researches demonstrate that SRO exhibits p-type conductivity, the authors should provide the Hall effect measurement of SRO film to confirm their argument.

Response: We thank the reviewer for finding the revision satisfactory. As per suggestion, we have performed a new experiment on newly prepared SRO thin film. Based on our experiment, we confirm that the SRO is n-type metal. We are grateful for this comment, which is now added in the supplementary Note.2.

Revision: In Supplementary Note 2.

“First, we have measured the carrier concentration type (n-or p-type) to draw the band structure. X-ray diffraction pattern (Supplementary Figure 8a) and longitudinal resistivity (Supplementary Figure 8c) confirmed the growth of SRO thin film on STO. From slope of the R_{xy} vs. out-of-plane magnetic field (μ_0H) (Supplementary Figure 8d), the carrier concentration is found to be $\sim 1.2 \times 10^{23} \text{ cm}^{-3}$ at room temperature. Negative slope is an indicative of n-type carriers in the SRO.

Supplementary Figure 8 | N-type metal SRO. a X-ray diffraction pattern of SRO (75nm)/STO. **b** Patterned Hall bar (6 μ m wide strip) of SRO (75 nm) along with the measurement circuit. **c** Longitudinal resistivity as a function of temperature; arrow indicates the ferromagnetic transition in SRO. **d** Measured transverse resistance (R_{xy}) as a function of out-of-plane magnetic field (± 8 T) at constant 1mA dc at 300K; inset shows the perpendicular magnetic anisotropy hysteresis measured at 50K. Line is fit to the data to calculate the carrier concentration at 300K.”

Reviewers' Comments:

Reviewer #3:

Remarks to the Author:

The authors have addressed all my comments satisfactorily. I recommend publishing this manuscript.